# Distributed Antenna in Drone Swarms: A Feasibility Study

Stuart William Harmer [1,*] and Gianluca De Novi [2]

1   School of Engineering, Computing & Mathematics, University of Chichester, Chichester PO19 6PE, UK
2   Harvard Medical School, Harvard University, Cambridge, MA 02138, USA
*   Correspondence: s.harmer@chi.ac.uk

**Abstract:** Unmanned aerial vehicles offer a versatile platform for the realization of phased array antenna systems, enabling multiple antenna elements to be distributed spatially in an agile, flexible, and cost-effective manner. Deploying individual antenna elements on single drones and using a swarm of such drones to create an antenna array has the potential to be a disruptive technology. Antenna directivity is limited by the physical aperture size as compared to the wavelength of the radiation being transmitted/received, with electrically larger antennas giving a higher directivity at the cost of an increased size and weight. The authors presented a brief feasibility study using a simple mathematical model implemented in software to explore the predicted performance of the novel UAV deployed antenna array, the limitations of such a system, and the potential applications where such a capability would be beneficial. The authors concluded that it is possible to achieve a suitably coherent superposition of electromagnetic radiation at frequencies of ~1 GHz and lower with current global positioning technologies which offer centimeter scale positioning accuracy and with current drone positioning systems used to control drone swarms.

**Keywords:** drone swarming; UAV; remote sensing; synthetic aperture; phased array; antenna; RADAR

## 1. Introduction

Unmanned aerial vehicles (UAVs), more commonly referred to as drones [1], are revolutionizing the aerospace industry by providing accessible and relatively inexpensive aerial platforms for an extremely wide variety of applications on a scale not realizable with traditional piloted aircrafts [2]. Initially, drones were deployed as replacements for tasks previously executed with manned aircrafts, for example, remote sensing applications [3], logistics [4], and civil defense [5]. However, one key differentiator of drones from traditional aircrafts is their capability to aerially distribute a large number of aircrafts simultaneously and to very accurately control their positions. Such formation flying is widely referred to as drone swarming and can be used for a wide variety of applications [6–10], and it has been effectively demonstrated for aerial displays, where drones are equipped with visible illumination sources and where they are flown in tightly controlled formations to produce light shows similar in effect to pyrotechnic displays [11]. It is entirely conceivable that the future applications of drone swarming will make use of its capability to fly multiple drones in a controlled formation for the collection or transmission of information and will potentially use it as a defense system.

Phased array antennas combine multiple spatially separated individual antenna elements to form a larger composite antenna, or synthetic aperture [12–17]. By controlling the timing of the transmission and/or reception of electromagnetic radiation, such an antenna is able to electronically beam steer with no physical change in orientation required for transmitting/receiving in different directions and to different focal planes. Phased array antennas operate by controlling the phase of the transmitted radiation when operated as a stepped frequency system or, equivalently, by altering the relative timing of the signals received/transmitted in the time domain [18]. Phased array antennas are widely used in a number of applications, including RADAR [19], communications [20], and remote





sensing [21]. The distribution of transceivers on multiple drones offers significant benefits due to the low cost, accessibility, and ease of operation that drones provide over more conventional piloted aircrafts. The authors herein term the antenna array composed of individual drones carrying antenna elements as a distributed antenna in drone swarming system, or the DADSS.

The motivation for this work was to establish a firm physical foundation for the discussion of the applications of an antenna array distributed over a drone swarm and, in particular, its beam forming capability, its sensitivity to the positional error of drones, and the number of drones required. With these basic parameters, this concept can be explored for specific applications where electrically large airborne antenna systems are advantageous and where it is impractical to deploy them with other technologies. Broadly, these areas are radar, remote sensing, communications, and defense.

## 2. Materials and Methods

A mathematical model of the DADSS's concept was developed by the authors and was implemented in the commercially available software MATLAB. The MATLAB model was used to briefly explore the potential performance of the DADSS, with emphasis on its novel ability to control beam patterns through distribution of the drone swarm and on the limitations imposed by positional uncertainty of the individual antenna elements carried by the drones.

We considered a swarm of drones (see Figure 1) which were located at positions $r_i$ and were controlled through adjustment of transmit phase to produce the desired constructive interference conditions of the transmitted/received EM field at position $R$. The EM field at a different position $r$ was of interest for determining the beam pattern produced by the array. The analysis presented was made by considering transmission of radiation in frequency steps, but the reciprocal nature of antenna ensured that the results also applied to received radiation. Symbols used in the following mathematical analysis are listed in Table 1.

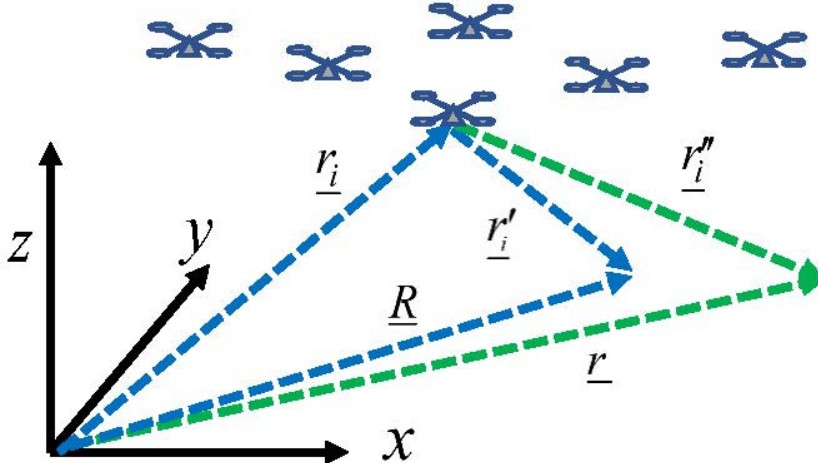

**Figure 1.** UAV swarm located with respect to a reference point located at the origin of the axes.

**Table 1.** Table of symbols used in mathematical analysis.

| Symbol | Quantity | Unit | Implemented in Model as |
|---|---|---|---|
| $r_i$ | Position of $i$th drone | m | Input |
| $r_i'$ | Vector displacement of steering position from $i$th drone | m | Calculated |
| $r_i''$ | Vector displacement of general position from $i$th drone | m | Calculated |
| $R$ | Steering position | m | Input |
| $r$ | General position | m | Input |
| $E_j$ | Electric field | $Vm^{-1}$ | Calculated |
| $c$ | Speed of light | $ms^{-1}$ | Input |
| $\mu_0$ | Permeability of free space | $Hm^{-1}$ | Input |
| $k_j$ | Wavenumber of $j$th frequency step | $m^{-1}$ | Calculated |
| $P_{i,j}$ | Power fed into the $i$th antenna element at the $j$th wavenumber | W | Calculated |
| $G_{i,j}$ | Gain of the $i$th antenna element at the $j$th wavenumber | - | Calculated |
| $\psi_{i,j}$ | Phase shift applied to the $i$th antenna element at the $j$th wavenumber. | Radians | Calculated |
| $b$ | Individual antenna side length | m | Input |
| $\theta_i$ | Angle between $i$th drone and vertical | Radians | Calculated |
| $I_j$ | Intensity of $j$th frequency step | $Wm^{-2}$ | Calculated |
| $f_j$ | Frequency of $j$th frequency step | Hz | Calculated |
| $f_c$ | Centre frequency | Hz | Input |
| $\sigma_f$ | Bandwidth | Hz | Input |
| $P_{Total}$ | Max power per element | W | Input |
| $\langle \phi_{i,j} \rangle$ | Mean phase of radiation from $i$th drone at $j$th frequency step | Radians | Calculated |
| $\sigma_j$ | Standard deviation of phase at $j$th frequency step | Radians | Input |
| $\Gamma_j$ | Array gain at $j$th frequency step | - | Calculated |
| $\Delta r$ | Standard deviation in position | m | Input |
| $\Delta t$ | Standard deviation in timing | s | Input |

The vectors in Figure 1 are related through Equations (1) and (2),

$$r_i + r_i' = R \qquad (1)$$

$$r_i + r_i'' = r \qquad (2)$$

and the magnitude of the electric field, $E_j\left(r\right)$, due to the radiation transmitted at the $j$th wavenumber, is given by Equation (3),

$$E_j\left(r\right) = \sqrt{\frac{\mu_0 c}{4\pi}} \sum_{i=1}^{N} \frac{\sqrt{P_{i,j}G_{i,j}}}{r_i''} \cos\left(k_j r_i'' + \psi_{i,j}\right) \qquad (3)$$

where $i$ labels the drone in the swarm and takes on integer values from 1 to $N$, $N$ being the total number of drones in the swarm; $j$ labels the wavenumber/frequency of transmission/reception of the antenna array and takes on integer values from 1 to $M$, $M$ being the number of frequency steps making up the transmitted/received signal; $\mu_0$, $c$, and $k_j$ are the permeability of free space, the speed of light, and the wavenumbers of the transmitted/received electromagnetic radiation, respectively; $P_{i,j}$ is the total power fed into the $i$th antenna element at the $j$th wavenumber; $G_{i,j}$ is the gain of the $i$th antenna element at the $j$th wavenumber; and $\psi_{i,j}$ is the phase shift which is applied to the $i$th antenna element at the $j$th wavenumber.

To steer the transmitted RF energy to constructively interfere with point $R$, we required that the phase of the radiation transmitted/received at each wavenumber from each antenna

in the array had the same phase at point *R*, and, without loss of generality, the phase at this wavenumber can be chosen to be zero; hence,

$$k_j r_i' + \psi_{i,j} = 0 \tag{4}$$

Equation (4) provides phase shifts $\psi_{i,j}$, which are required to 'beam steer' to point *R*, and, hence, by applying Equations (1)–(4), the electric field, due to the radiation transmitted at the *j*th wavenumber, can be written as Equation (5),

$$E_j(r) = \sqrt{\frac{\mu_0 c}{4\pi}} \sum_{i=1}^{N} \frac{\sqrt{P_{i,j} G_{i,j}}}{\left\| r - r_i \right\|} \cos\left( k_j \left( \left\| r - r_i \right\| - \left\| R - r_i \right\| \right) \right) \tag{5}$$

The intensity of the electromagnetic field due to the radiation transmitted at the *j*th wavenumber can be obtained from Equation (5) as follows,

$$I_j(r) = \frac{E^2}{\mu_0 c} = \frac{1}{4\pi} \left( \sum_{i=1}^{N} \frac{\sqrt{P_{i,j} G_{i,j}}}{\left\| r - r_i \right\|} \cos\left( k_j \left( \left\| r - r_i \right\| - \left\| R - r_i \right\| \right) \right) \right)^2 \tag{6}$$

Equation (6) gives the intensity of the electromagnetic radiation due to the radiation transmitted at the *j*th wavenumber at point *r*. Each antenna element carried by a UAV was treated here as being an identical square antenna with side length *b*, and all the UAVs were assumed, for simplicity, to adopt the same orientation with antenna elements pointing toward the ground. Thus, each antenna element has a modelled gain function given by Equation (7),

$$G_{i,j} = \frac{(k_j b)^2}{\pi} \frac{\sin^2\left( \frac{k_j \theta_i b}{2} \right)}{\left( \frac{k_j \theta_i b}{2} \right)^2} \tag{7}$$

where angle $\theta_i$ is the angle made between $r_i''$ and the z-axis (see Figure 1).

Critically, lack of accurate knowledge on position of the UAVs comprising the array had a detrimental effect on the direction of EM radiation into the target region. When letting positions $r_i$ be normally distributed with mean of $r_i$ and standard deviation $\Delta r$, by virtue of the effect of generating inaccurate $\psi_{i,j}$, there occurs a degradation in steering of the beam to the desired location as prescribed in Equations (5) and (6). In addition to uncertainty in UAV positions, which was limited by the accuracy of GNSS systems and by positional control of the UAVs comprising the DADSS, there was also error in phase $\psi_{i,j}$ due to inability to precisely control the phase/timing of the transceiver elements while letting the standard deviation of timing be $\Delta t$. The corresponding phase was also a normally distributed random variable with mean $\langle \phi_{i,j} \rangle$ and standard deviation $\sigma_j$ as given by Equations (8) and (9), respectively,

$$\langle \phi_{i,j} \rangle = k_j \left( \left\| r - r_i \right\| - \left\| R - r_i \right\| \right) \tag{8}$$

$$\sigma_j = k_j \sqrt{\Delta r^2 + (c \Delta t)^2} \tag{9}$$

where the standard deviation of position of the UAV is assumed to be $\Delta r$ meters; the standard deviation of timing is assumed to be $\Delta t$ seconds; and, for simplicity, these are the same for each UAV that comprises the array. The effects of transmitted/received

bandwidth were included by calculating the spatial distributions of Equation (6) over a range of uniformly spaced wavenumbers according to Equation (10),

$$k_j = k_1 + (j-1)\Delta k \tag{10}$$

where $k_1$ is the lowest wavenumber, $\Delta k$ is the wavenumber step size, and $j$ runs from 1 to $M$. The wavenumber is related to the frequency, $f$, in Hz, of electromagnetic radiation through

$$k = \frac{2\pi f}{c} \tag{11}$$

Radiation was transmitted from each UAV element in frequency steps as defined by Equations (10) and (11), with each frequency step having a radiated power which was assumed to be Gaussian in power density $\rho_j$, in WHz$^{-1}$, and with a total power per element of $P_{Total}$, in Watts, such that

$$\rho_j = \frac{P_{Total}}{\Delta f \sum\limits_{j=1}^{M} e^{-\frac{1}{2}\left(\frac{f_j - f_c}{\sigma_f}\right)^2}} e^{-\frac{1}{2}\left(\frac{f_j - f_c}{\sigma_f}\right)^2} \tag{12}$$

where $f_c$ is center frequency of the pulse, $\sigma_f$ is the bandwidth, and $\Delta f = \frac{c}{2\pi}\Delta k$ is the frequency step size. Accordingly, the power fed to each antenna element at frequency step $j$ was obtained from Equation (12) to give Equation (13),

$$P_{i,j} = \rho_j \Delta f = \frac{P_{Total}}{\sum\limits_{j=1}^{M} e^{-\frac{1}{2}\left(\frac{f_j - f_c}{\sigma_f}\right)^2}} e^{-\frac{1}{2}\left(\frac{f_j - f_c}{\sigma_f}\right)^2} \tag{13}$$

The gain of the antenna array at the $j$th wavenumber was then calculated as

$$\Gamma_j(r) = \frac{4\pi I_j(r)}{\sum\limits_{i=1}^{N} \frac{P_{i,j}}{\left(r_i''\right)^2}} \tag{14}$$

and, consequently, the mean (averaged over wavenumbers) gain of the antenna array was obtained from Equation (14) by summing wavenumbers and dividing by the number of bands, $M$,

$$\left\langle \Gamma(r) \right\rangle = \frac{4\pi}{M} \sum\limits_{j=1}^{M} \left( \frac{I_j(r)}{\sum\limits_{i=1}^{N} \frac{P_{i,j}}{\left(r_i''\right)^2}} \right) \tag{15}$$

Note that, in the case of antenna elements that have no losses, such as those in Equation (7), the antenna array mean gain, Equation (15), is equal to the array directivity. For the sake of simplicity, we assumed lossless antennas in this study and accepted that the results would, therefore, somewhat overestimate real world performance.

The distribution of drone-carried antenna elements was important in determining the beam pattern in the plane of focus of the DADSS, with drones that could be randomly distributed over volumes, surfaces, or paths as determined by the desired beam pattern. Drones had to be distributed in a random manner in order to mitigate grating lobes, where the beam had multiple significant lobes in different directions from the array. Grating lobes arise because of constructive interference from periodically spaced sources; random positioning of the individual antenna elements allows the antenna element spacing to

exceed the usual half-wavelength spacing constraint for phased arrays with periodically spaced elements. Periodically spaced antenna elements result in grating lobes if the spacing is greater than the half-wavelength limit, and this was a highly undesirable constraint for the DADSS, where drone separation was hard-limited by the size of the drone. In the presented model, the number of drones in the swarm (and, hence, the number of antenna elements) was specified, as were the position and size of the cuboid that drones were to be randomly distributed within. Alternatively, a surface could be chosen over which the drones could be randomly distributed, or a path could be chosen along which the drones could be randomly located. Due to the random placement of the drones, physical quantities calculated by the model varied from run to run, even when the model parameters remained fixed. In addition to the drone distribution, the model had the following user-defined inputs: number of drones comprising the DADSS; steering coordinates of beam; grid shape, size, position, and spatial resolution over which the electric field, intensity, and gain distributions were calculated; center frequency; frequency step size and bandwidth of the pulse; size of the individual drone-carried antenna elements; and standard deviation of drone position errors (assumed to be normally distributed with mean at desired position and standard deviation set as a user-defined parameter).

## 3. Results

Of foremost interest for evaluating the potential performance of the DADSS was the directivity of the antenna array that could be realized from the drone swarm, as this measures its capability to transmit/receive radiation in the desired direction, and the beam pattern of the antenna, as this determines the shape of the region into which or from which the radiation is transmitted or received.

The calculation of the gain, Equation (15), provided this information and was presented graphically for some selected cases with a DADSS that comprised 24 drones, which were randomly distributed among swarms that were either cuboids of different dimensions, specified surfaces, or paths.

Figure 2 shows the randomly distributed locations of the 24 drones within $25 \times 5 \times 1 \text{ m}^3$ that comprised the DADSSs. Each drone within a swarm carried a 0.3 m diameter antenna which transmitted/received radiation; the gain of an individual antenna was 12.6 at a center frequency of 1 GHz, and this can be contrasted with the much larger gains that were predicted with multiple drones. The power spectrum of the transmitted electromagnetic radiation is shown graphically in Figure 3. Figures 4–7 all give the results of the DADSSs with this same power spectrum. In addition to the graphical plots of the antenna gain in some specific cases, Figures 4–7 and Tables 2 and 3 provide the data obtained from running a variety of different scenarios 100 times each and from calculating the mean maximum gain and the mean of the beam widths as measured to the first $-3$ dB points of the resulting antenna patterns. Since the DADSSs presented were all electrically large, the far-field region was located beyond a significant range from the antenna. The directivity of the DADSSs could be measured at different ranges, which were either within the far-field region or near-field region according to Equation (15), and these are also tabulated in Tables 2 and 3.

$$r_{FF} \sim \frac{fL^2}{c} \tag{16}$$

where $L$ is the largest dimension of the array.

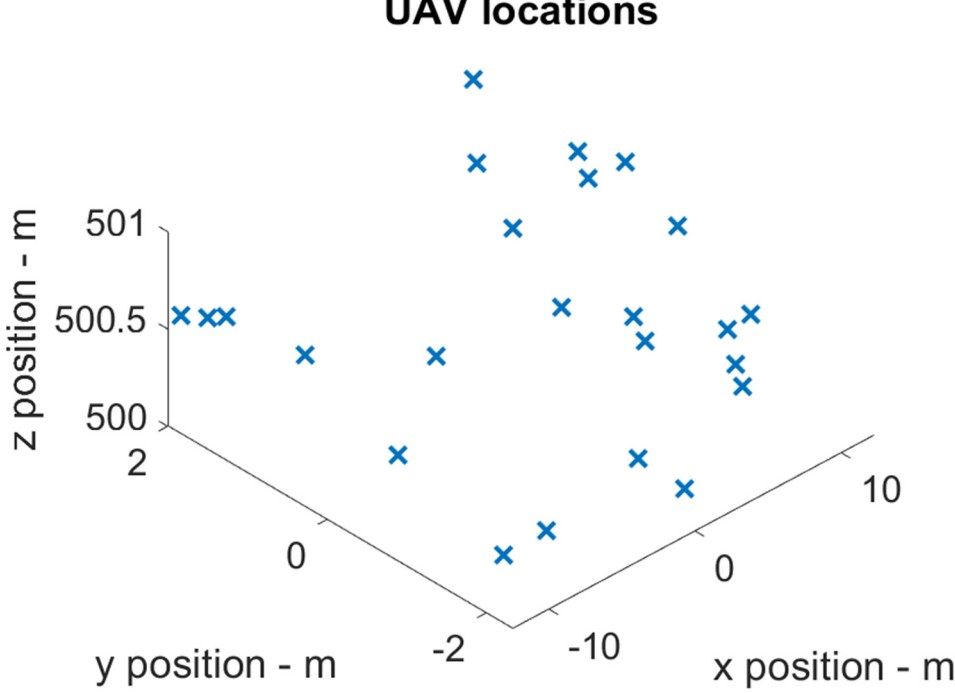

**Figure 2.** Drone positioning for the DADSSs comprising 24 drones, which were randomly distributed within a cuboid of 25 × 5 × 1 (x, y, z) m.

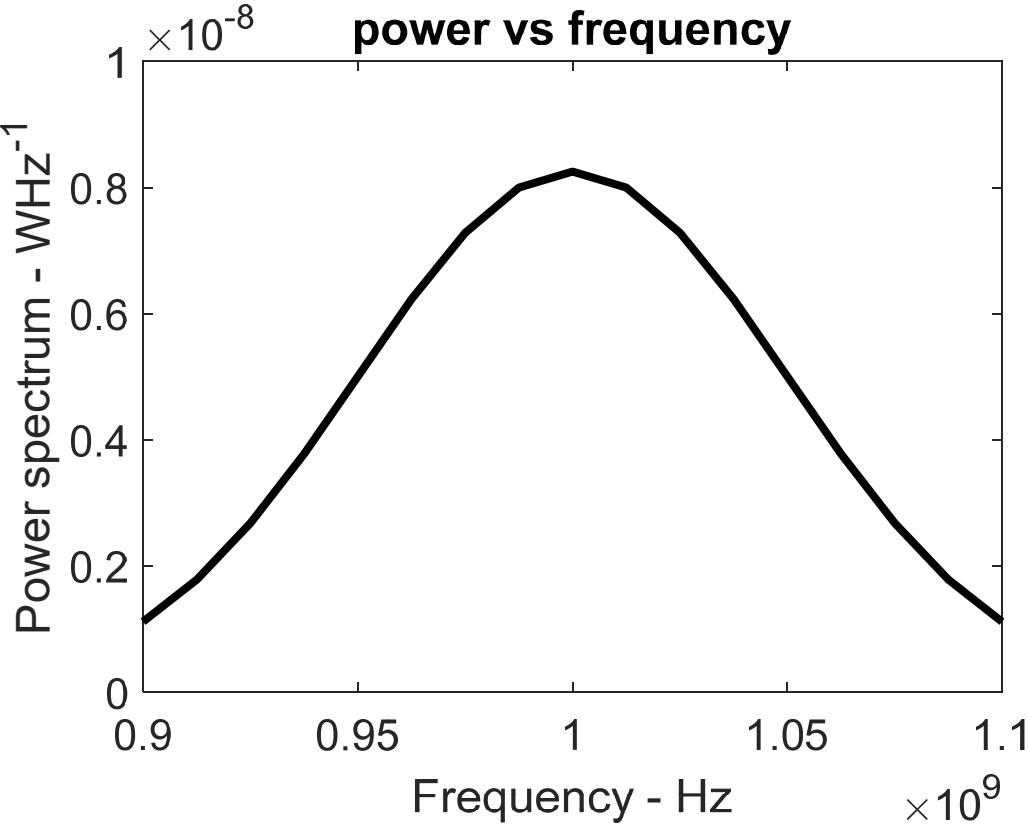

**Figure 3.** Gaussian power spectrum of the transmitted pulse for Figures 4–7: the pulse had a center frequency of 1 GHz and pulse width of 50 MHz.

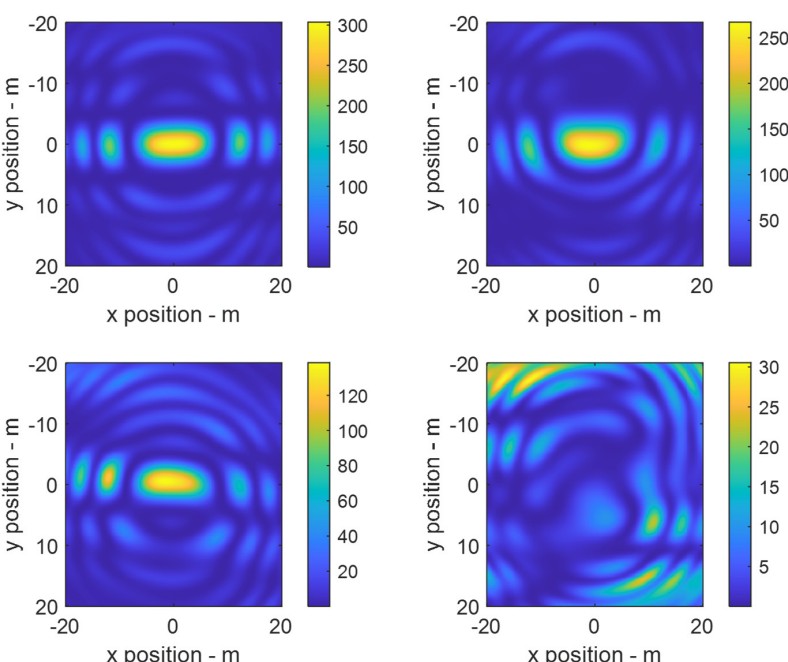

**Figure 4.** Effect of positional uncertainty. Predicted antenna gain patterns of DADSSs comprising 24 drones, which were randomly distributed within $25 \times 5 \times 1$ (x, y, z) $m^3$. The DADSSs were steering at a point that was directly 500 m below the drone swarm. In the top left figure, there is no error in drone positioning; in the top right figure, the error in drone positioning is 2 cm; in the bottom left figure, the error is 5 cm; and, in the bottom right figure, the error is 10 cm. Note the deformation of the beam shape and drop in gain with increasing error in drone positioning. See color bars associated with each individual plot to determine gain.

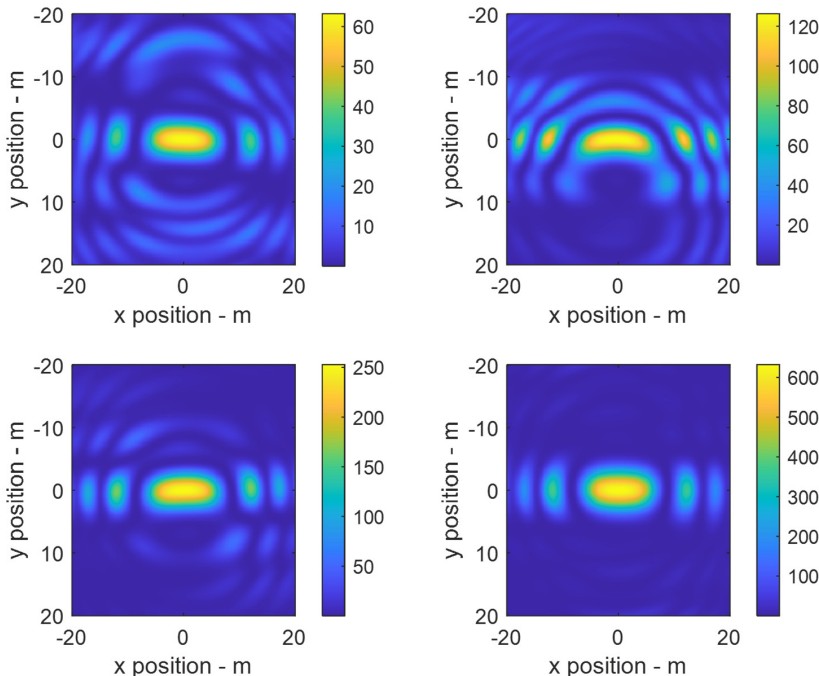

**Figure 5.** Effect of number of drones. Predicted antenna gain patterns of DADSSs comprising 5 drones (**top left**), 10 drones (**top right**), 20 drones (**bottom left**), and 50 drones (**bottom right**), which were randomly distributed within $25 \times 5 \times 1$ (x, y, z) $m^3$. The DADSSs were steering at a point that was directly 500 m below the drone swarm. Increase in number of drones increased gain and improved symmetry of desired beam pattern.

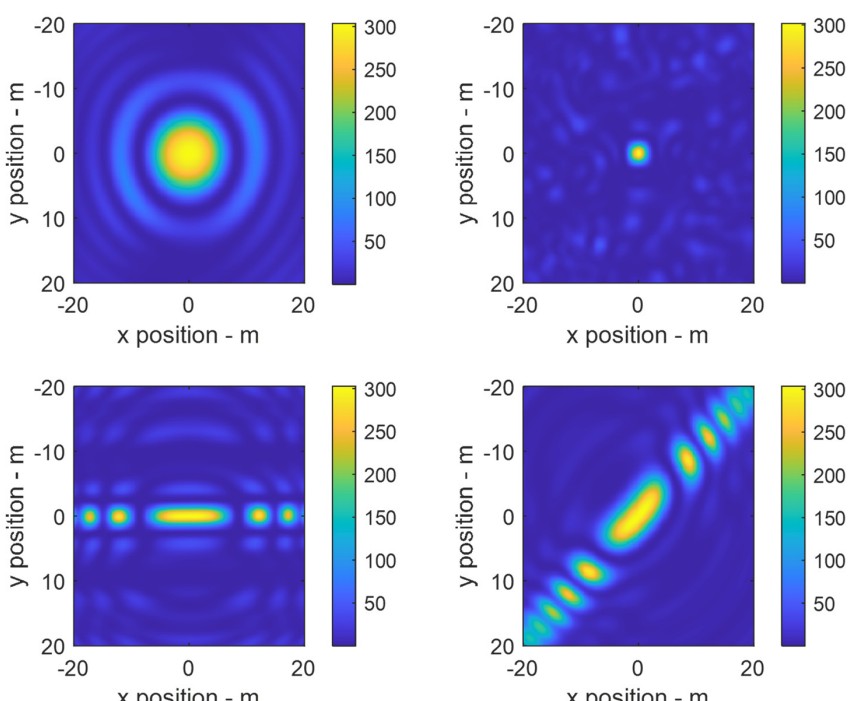

**Figure 6.** Effect of drone distribution. Antenna gain pattern of DADSSs comprising 24 drones, which were distributed randomly within $10 \times 10 \times 1$ (x, y, z) m$^3$ (**top left**), within $50 \times 50 \times 1$ m$^3$ (**top right**), within $50 \times 1 \times 1$ m$^3$ (**bottom left**), and along the line y = x over values of $-10 < \times < 10$ m (**bottom right**). The DADSSs were steering at a point that was directly 500 m below the drone swarm. Control of the resulting beam pattern was realized by changing the distribution of drones in the swarm to give apertures of the desired shape.

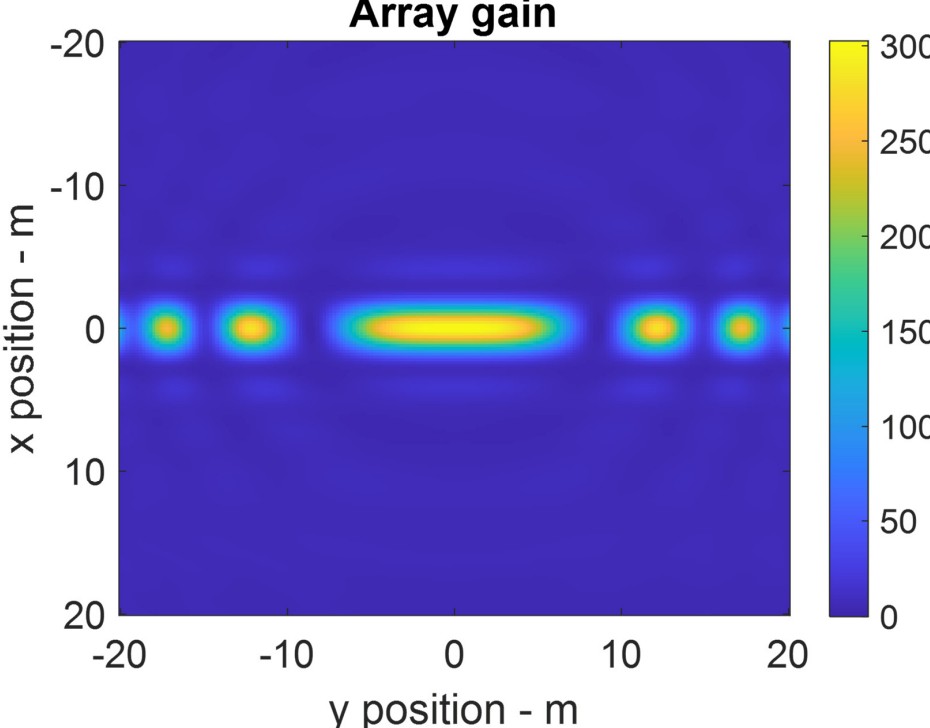

**Figure 7.** Spatial averaging. Beam gain pattern averaged over 100 runs for a DADSS consisting of 24 drones, which were randomly distributed within $50 \times 1 \times 1$ m$^3$. The DADSS was steering at a point that was directly 500 m below the drone swarm.

**Table 2.** Table of beam pattern data for a variety of model parameters of a swarm of 24 drones, which were equipped with 0.3 m antennas, were operating at a center frequency of 1 GHz, and were located 500 m from focus plane. The beam size at 500 m was measured to the first −3 dB points in the pattern.

| Number of Drones | Mean Maximum Gain | −3 dB Mean Beam Size at 500 Meters m | Field Region at 500 m | Drone Position Uncertainty m | Distribution |
|---|---|---|---|---|---|
| 24 | 302 | Width 2.34 Length 11.4 | Near | 0 | Cuboid 50 × 5 × 1 m |
| 24 | 112 | Width 2.60 Length 11.8 | Near | 0.05 | Cuboid 50 × 5 × 1 m |
| 24 | 302 | Width 4.91 Length 4.98 | Near | 0 | Cuboid 25 × 25 × 1 m |
| 24 | 109 | Width 4.75 Length 5.06 | Near | 0.05 | Cuboid 25 × 25 × 1 m |
| 24 | 303 | Width 9.79 Length 9.95 | Far | 0 | Cuboid 10 × 10 × 1 m |
| 24 | 109 | Width 9.70 Length 9.56 | Far | 0.05 | Cuboid 10 × 10 × 1 m |
| 24 | 296 | Width 1.23 Length 1.25 | Near | 0 | Cuboid 100 × 100 × 1 m |
| 24 | 105 | Width 1.26 Length 1.27 | Near | 0.05 | Cuboid 100 × 100 × 1 m |
| 24 | 109 | Width 5.30 Length 11.32 | Far | 0.05 | The line y = x, distributed between values −10 < x < 10 and −10 < y < 10 m |
| 24 | 110 | Width 6.42 Length 6.38 | Far | 0.05 | The surface of a spherical cap, with radius of 500 m and with x and y distributed between −10 and 10 m |

**Table 3.** Table of beam pattern data for a variety of model parameters of swarms comprising different numbers of drones, which were equipped with 0.3 m antenna, were operating at a center frequency of 1 GHz, and were located 500 m from focus plane. The beam size at 500 m was measured to the first −3 dB points in the pattern.

| Number of Drones | Mean Maximum Gain dB | −3 dB Mean Beam Size at 500 Meters m | Field Region at 500 m | Drone Position Uncertainty m | Distribution |
|---|---|---|---|---|---|
| 5 | 56 | Length 1.67 Width 1.60 | Near | 0.02 | Cuboid 100 × 100 × 1 m |
| 10 | 105 | Length 1.37 Width 1.42 | Near | 0.02 | Cuboid 100 × 100 × 1 m |
| 25 | 262 | Width 1.28 Length 1.30 | Near | 0.02 | Cuboid 100 × 100 × 1 m |
| 50 | 521 | Width 1.29 Length 1.29 | Near | 0.02 | Cuboid 100 × 100 × 1 m |
| 100 | 1040 | Width 1.29 Length 1.29 | Near | 0.02 | Cuboid 100 × 100 × 1 m |
| 250 | 2580 | Width 1.28 Length 1.28 | Near | 0.02 | Cuboid 100 × 100 × 1 m |

Since the effective antenna array sizes could be many times the wavelength of the transmitted/received radiation, the far field ranges may be significant, and the DADSSs which had a beam form in the near-field region, i.e., less than the range given in Equation (16),

were envisioned for remote ground sensing applications, electronic warfare applications, and for communications. As an example, for a DADSS distributed within a cuboid with a maximum dimension of 50 m operating at a center frequency of 1 GHz, the far-field distance was ~9 km; whereas for a DADSS distributed in a cuboid with a maximum dimension of 10 m operating at a center frequency of 100 MHz, the far-field distance was ~30 m.

The positional uncertainty was critical for beam forming in the DADSSs, with the requirement being that the positional uncertainty must be significantly less than the center frequency wavelength of the transmitted/received radiation, i.e., $\Delta r << \lambda$. The effect of the positional uncertainty of the drones is evident in Figure 4, where increasing the standard deviation of the drone position degraded the beam pattern and reduced the maximum gain. Operating at higher frequencies and, hence, shorter wavelengths required greater precision in order to achieve good beam patterns with a high directivity. Hence, the proposed concept was most easily exploited out of doors for longer wavelengths, where the frequency was ≤1 GHz. The standard deviations used to produce Figure 4, when expressed as a percentage of the wavelength of transmission/reception, were 0% for the top left plot, 7% for the top right plot, 17% for the bottom left plot, and 33% for the bottom right plot. An uncertainty at or below 5% of the wavelength range was acceptable, with minimal degradation in the beam shape and loss of gain. The number of drones was also important when forming a highly directive antenna and when producing predictable beam patterns at the focus plane. Figure 5 shows the effect of increasing the number of drones from 5 in the top left plot to 50 in the bottom right plot. This effect is clearly seen in Figure 5, with a ten-fold increase in the drones resulting in a similar increase in the maximum gain. However, in addition to the maximum gain, the beam pattern quality was also significantly affected by the number of drones, with the randomization of small numbers of drones resulting in a higher asymmetry of the beam pattern than large numbers of drones. This effect was explained by the higher asymmetry in the radiation sources (the drones) that made up the aperture when there were fewer randomly located drones. Numbers exceeding 20 drones were usually required to ensure a predictable radiation pattern, and, for the purposes of this paper, 24 drones were considered to be sufficient for providing reliable beam patterns and practical from a deployable system aspect. Finally, the formation of the randomly located drones was utilized to provide an optimal beam pattern for the application. Since the drones formed an aperture and since the beam pattern was dependent upon the shape of this aperture, beam shaping could be achieved through the suitable distribution of the drones within the swarm. The distribution could be within a volume (3D), over a surface (2D), or along a path (1D). Figure 6 shows the beam patterns resulting from a swarm of 24 drones that were arranged within a small $(10 \times 10 \times 1 \text{ m})^3$ cuboid (top left figure), within a larger $(100 \times 100 \times 1 \text{ m}^3)$ cuboid (top right figure), within a long and thin $(50 \times 1 \times 1 \text{ m}^3)$ cuboid (bottom left figure), and along the line y = x to produce an angled beam pattern (bottom right figure). Although simple distribution schemes were chosen for Figure 5, the ability to control the beam pattern at the desired plane was clearly evident, and it was entirely possible to match the distribution of the drone swarm to the required beam pattern for an application by acquiring the information dynamically and then modifying the distribution of the drone swarm accordingly. Since the drones were distributed randomly, running the same model parameters gave somewhat different beam patterns on each run. The variation was greatest for small numbers of drones and becomes less notable for significant numbers of drones. These effects could be mitigated by finding the mean spatial distribution of the beam patterns for a given set of model parameters. This was done by averaging over 100 runs, with the key figures of merit, maximum gain, and beam widths recorded and tabulated in Tables 2 and 3. An example of a beam pattern averaged over 100 runs is provided in Figure 7.

## 4. Discussion

Antennas are ubiquitous, being present in every device that utilizes electromagnetic radiation to communicate or collect information over a channel [22]. As such, any antenna

technology which is disruptive is required to offer a performance and capability which is significantly beyond that of the current antenna technologies. The DADSS can provide the capability of conventional phased array antennas, which include electronic beam steering with the new and novel ability of the antennas being able to three-dimensionally morph the aperture shape by simply moving the drone-controlled elements into different positions. Other novel properties include being able to add or remove elements from the antennas and being able to form, undertake the desired task, and then disperse the antenna array, which is similar to one of the key desirable characteristics of plasma antennas [23] and which would be a desirable attribute for most defense applications.

Forming electronically steerable antenna arrays using spatially distributed antenna elements carried on drones has been suggested by others [24–27] and has numerous potential applications; these include the use of airborne antennas for the remote sensing of ground water [28], geologic mapping [29], tomography [30], humanitarian demining [31], the detection and location of buried improvised explosive devices (IEDs) [32], mineral and oil exploration and surveying [33], and electronic warfare [34]. The DADSS can provide beam sizes which are completely unachievable using conventional airborne antennas, for example, an antenna carried by a helicopter or a single UAV, and its resolution is comparable to the track SAR resolution [35]. However, with SAR, the resolution is enhanced only in the direction of travel, whereas the DADSS can provide an excellent spatial resolution in both lateral beam directions because of the large effective antenna apertures that are attainable with the drone swarm and its possible operation in the near-field region of the antennas. Operating at a center frequency of 1 GHz, a DADSS system comprised of 24 drones, which are randomly dispersed within a $100 \times 100$ m square and which flying at an altitude of 500 m, can be distributed to give a beam width of ~1 m. The flexibility of being able to deploy an electrically large antenna with a reconfigurable beam pattern allied with ease of deployment and the capability to operate over any terrain offers a disruptive antenna technology for remote sensing, communications, and defense applications.

The current GNSS systems can provide a positioning accuracy at the centimeter scale [36], and the technologies needed to construct phased array antennas operating at ~1 GHz are readily available. UAV technology has advanced rapidly over the last decade, with inexpensive and high-performance off-the-shelf drones now available that could act as the antenna element carriers. The software required to control the drone swarm to realize the DADSS concept would be specialized to enable the randomization of the drones within the predefined space and might also incorporate artificial intelligence to enable rapid and dynamic beam optimization. However, the control of multiple drones within a swarm has been amply demonstrated [37]. The timing/phasing of the array elements in an antenna system that can alter the separation between elements provides, perhaps, the greatest technical challenge.

The authors' contribution to the nascent field of distributed antenna arrays formed using drone swarming was primarily in predicting the beam patterns achievable with such antenna and the sensitivity of such patterns to positional errors, the number of drones in swarm, and the distribution of the drones within the swarm. The authors concluded that the teaming of multiple drones to achieve the constructive interreference of electromagnetic and, potentially, acoustic waves is entirely feasible with the current technology, offers a disruptive performance for airborne antennas, and has many potential applications.

## 5. Patents

A patent protecting the DADSS concept was filed on 24 February 2022.

U.S. Patent Application No.: 17/679,755Filing Date: 24 February 2022

Title: Mobile Reconfigurable Distributed Aperture Systems and Methods for Remote Sensing

Applicant: Plymouth Rock Technologies Inc.

Inventors: Harmer et al.

**Author Contributions:** Conceptualization, S.W.H. and G.D.N.; methodology, S.W.H.; software, S.W.H.; validation, S.W.H. and G.D.N.; formal analysis, S.W.H.; writing—original draft preparation, S.W.H.; writing—review and editing, S.W.H. and G.D.N. All authors have read and agreed to the published version of the manuscript.

**Funding:** This research received no external funding.

**Institutional Review Board Statement:** Not applicable.

**Informed Consent Statement:** Not applicable.

**Data Availability Statement:** Not applicable.

**Conflicts of Interest:** The authors declare no conflict of interest.

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
