# Peer review of "Distributed Antenna in Drone Swarms: A Feasibility Study"

_drones, doi:10.3390/drones7020126_

Round 1

Reviewer 1 Report

Please see the attached comments.

Author Response

The authors thank the reviewer for their time in reviewing the article and for their excellent recommendations which will certainly improve the article. The authors have made the following alterations in light of the review:

  1. The authors have significantly revised the abstract in line with the reviewer’s recommendations and suggestions (response to point 1)
  2. The authors have removed the subjective statement (response to point 2) from the introduction
  3. The authors acknowledge and cite some examples of other work on using single and swarmed UAV for synthetic aperture/ distributed antenna systems. See references [24-27]. (response to point 3)
  1. The authors have included the suggested references and thank the reviewer for bringing these articles to their attention. See references [15-17]. (response to point 4)
  2. The authors have added motivations for the work at the end of the introduction section and the contributions of the paper at the end of the discussion section. (response to point 5)
  3. No, the drones are distributed randomly in 3D space. The model allows the random distribution of drones within a defined cuboid, over a defined surface or along a defined path. The positions are referenced with respect to some arbitrary location (the origin in Figure 1.). Once all positions of drones relative to this point are known (there is an uncertainty in this knowledge) the phase shifts required for each drone and at each frequency step can be calculated with respect to an arbitrary drone in the swarm (which has no phase shift applied and serves as the reference). (response to point 6)
  4. The authors have added a table of symbols (Table 1) as suggested and linked these to the code used in the simulation. (response to point 7)

Reviewer 2 Report

This is an interesting work with great potential for applicability. The methodological design is clear and has data that allow us to agree with the arguments put forward.

I did not find elements that disagree with the data presented, especially in figures 3, 4 and 5 and in tables 1 and 2.

All data are explored and evaluated in the discussion, which is well argued.

The presented results encourage to innovate with the discussed technology. This is an interesting work with great potential for applicability. The methodological design is clear and has data that allow us to agree with the arguments presented.

I did not find elements that disagreed with the data presented, especially in figures 3, 4 and 5 and in tables 1 and 2.

The discussion is well argued, with all data explored and evaluated.

The presented results encourage to innovate with the discussed technology. By deploying DADSS , drones will be able to be used on a large scale in sectors of the economy such as agriculture (spraying or monitoring). 

Author Response

The authors than the reviewer for their time and positive comments or their work and note that some minor revisions have been made, which the authors believe improve the article. 

Reviewer 3 Report

Dear colleagues, in theory your work is very good, but I have a few comments that you did not cover clearly in your paper. As a researcher with drone piloting experience, I am not entirely sure about the practical feasibility of your theory/model. When the drones (mentioned in paper) fly together in close proximity, the wind gusts/turbulences generated by the rotors make it impossible to keep them in the exact position (I fly with RTK drones, with many years of experience) - so the risk of collision is enormous. At an flying altitude of 500m AGL (or whatever) e.g. the control/visibility of the drones is already limited: what sensors would they entrust to the flight together in the swarm? The reflective (antenna?) surface of the drone (mounted on it) can really disturb the flight, it can have a special deflecting effect in the flight. What size/what type of UAV (perhaps: UAS) systems were you thinking about?

A few technical notes:

- In figure 1, the origin of the coordinate axes does not coincide, write axes (there is only Z), the green arrows merge

- Figure 2 needs to be improved: the network of the coordinate system (in the x/y direction) so that the height difference in the Z direction can also be seen, or rotate the 3D diagram, or show the position of the drones in 2 (2D) diagrams from a top view and a side view .

Author Response

The authors thank the reviewer for their time and constructive recommendations. The reviewer’s questions are indeed valid and there are some significant technical hurdles to realize the system. The drones would need to be controlled autonomously and their positions measured an updated continuously using GNSS, this position information would be used to control the drone to try to place it close to the required calculated position within the swarm. The effect of errors in positions is important and is covered in the paper, but atmospheric effects that may limit positional control accuracy are not considered. These would indeed be limiting as the reviewer points out. So, such a system would be degraded or rendered useless in windy conditions or conditions where errors in position exceed about 1/3 – ½ wavelength. Hence the system becomes more realistic with longer wavelength RF. The authors have not included details of the type of drone considered, as the study’s goal was to explore the basic principle and effects. However, the authors would expect relatively large drones with and ability to hover (i.e. helicopter drones or VTOL fixed wing machines). The payload capacity would also need to be carefully considered and would be dependent on the application and type of RF transceiver equipment required.

The authors respond to the reviewers technical notes below (response in italics)

- In figure 1, the origin of the coordinate axes does not coincide, write axes (there is only Z), the green arrows merge

The z axis (black) is used for altitude. The authors have added x, and y for clarity. The green and blue lines indicate the vector positions of drones within the swarm (relative to the origin) and also the steering position for the radiation.

- Figure 2 needs to be improved: the network of the coordinate system (in the x/y direction) so that the height difference in the Z direction can also be seen, or rotate the 3D diagram, or show the position of the drones in 2 (2D) diagrams from a top view and a side view .

Figure 2 is really only included to show the reader an output of the code, which in this case is the random allocation of 24 drones within a 25 by 5 by 1 metre cuboid. In reality, drones cannot be placed arbitrarily close to one another because of the drones finite size and, as the reviewer points out, the possibility of collision.

Round 2

Reviewer 1 Report

The authors have addressed my concerns, no further comments.